# CAVE: Connectome Annotation Versioning Engine

Sven Dorkenwald [1,2,10], Casey M. Schneider-Mizell [3,10], Derrick Brittain[3], Akhilesh Halageri[1], Chris Jordan[1], Nico Kemnitz [1], Manual A. Castro[1], William Silversmith[1], Jeremy Maitin-Shephard[4], Jakob Troidl[5], Hanspeter Pfister [5], Valentin Gillet [6], Daniel Xenes[7], J. Alexander Bae [1,8], Agnes L. Bodor[3], JoAnn Buchanan [3], Daniel J. Bumbarger[3], Leila Elabbady[3], Zhen Jia[1,2], Daniel Kapner[3], Sam Kinn[3], Kisuk Lee[1,9], Kai Li [2], Ran Lu[1], Thomas Macrina[1,2], Gayathri Mahalingam[3], Eric Mitchell [1], Shanka Subhra Mondal [1,8], Shang Mu[1], Barak Nehoran [1,2], Sergiy Popovych[1,2], Marc Takeno [3], Russel Torres [3], Nicholas L. Turner[1,2], William Wong[1], Jingpeng Wu[1], Wenjing Yin[3], Szi-chieh Yu [1], R. Clay Reid [3], Nuno Maçarico da Costa [3], H. Sebastian Seung [1,2] & Forrest Collman [3] ✉

Advances in electron microscopy, image segmentation and computational infrastructure have given rise to large-scale and richly annotated connectomic datasets, which are increasingly shared across communities. To enable collaboration, users need to be able to concurrently create annotations and correct errors in the automated segmentation by proofreading. In large datasets, every proofreading edit relabels cell identities of millions of voxels and thousands of annotations like synapses. For analysis, users require immediate and reproducible access to this changing and expanding data landscape. Here we present the Connectome Annotation Versioning Engine (CAVE), a computational infrastructure that provides scalable solutions for proofreading and flexible annotation support for fast analysis queries at arbitrary time points. Deployed as a suite of web services, CAVE empowers distributed communities to perform reproducible connectome analysis in up to petascale datasets (~1 mm³) while proofreading and annotating is ongoing.

Volume electron microscopy (EM) provides an exquisite view into the structure of neural circuitry and is currently the only technique capable of reconstructing all synaptic connections in a block of brain tissue. EM imagery not only facilitates the reconstruction of neuronal circuits but also enables scientists to combine them with rich ultrastructure visible in these images[1–6] (Fig. 1a). An increasing set of ultrastructural features, such as synapses[1,2,7,8], their neurotransmitter identity[2,9] and mitochondria[1–5], can be extracted automatically through machine learning methods. In addition, human experts have long used EM to make a rich set of observations about cellular and subcellular processes, including the

localization of a wide range of organelles and cell-to-cell interactions[10,11]. When combined with neuronal reconstructions, these datasets enable analyses of richly annotated connectomes[12–18]. This trend is mirrored in other data-intensive fields such as genome sequencing[19] and large-scale astronomy surveys[20], where raw data are iteratively enriched as increasingly accurate and diverse sets of annotations are added.

Today, neurons in EM datasets are extracted through automated approaches[21–24] to scale analyses to increasingly larger volumes[12,14–16]. However, manual proofreading of automated segments is still necessary to achieve reconstructions suited for analysis[25]. Proofreading of

**Fig. 1 | Proofreading and analysis of connectomics datasets. a**, A rich set of ultrastructural features can be extracted from EM images and used for analysis. The corresponding ultrastructural features are annotated with a red asterisk (*). The synapse is annotated with a red arrow pointing from the presynaptic site to the postsynaptic site. **b**, Large connectomics datasets are proofread, annotated and analyzed by a distributed pool of users in parallel. **c**, Proofreading adds and removes fragments from cell segments (left, before proofreading; center, removed and added fragments; right, after proofreading). **d**, Synapse assignments have to be updated with proofreading. All synapses (within the cutout) that were added and removed though the proofreading process of the cell in **c** are shown. Scale bars, 100 μm (**c**), 1 μm (**a**: synapse, mitochondria), 10 μm (**a**: nuclei) and 20 μm (**d**). *T*, time.

large datasets takes years, but even partial connectomic reconstructions produced along the way are useful for analysis. This raises the need for software infrastructure that facilitates concurrent proofreading within large collaborations or entire communities of scientists and proofreaders each working on their individual analyses (Fig. 1b). However, existing tools and workflows only support static exports after proofreading has been completed[26,27].

To enable this shift to collaborative proofreading and analysis of connectomics datasets, we created the Connectome Annotation Versioning Engine (CAVE). Although we developed CAVE for connectomics datasets, it introduces methods and concepts that are generally applicable to other fields that face similar challenges. Within connectomics, CAVE is generally applicable to all connectomics datasets produced with different microscopy methods, segmentation and sizes, including petascale datasets. Imaged at a resolution of ~10 nm, the raw imagery of these ~1-mm³-sized datasets takes up ~1 petabyte (refs. 14,15).

For proofreading, CAVE builds on the ChunkedGraph[28,29]. Like previous systems[26,30,31], the ChunkedGraph represents cells as connected components in a supervoxel (groups of voxels) graph. It is currently the only system for neuron-based proofreading by a distributed community but was too costly to be used on petascale datasets. Here, as one part of CAVE, we introduce the next generation of this system, the ChunkedGraph v2, which scales proofreading to petascale datasets through a more cost-efficient storage implementation. Ongoing proofreading presents a challenge for analysis: edits not only add and remove fragments from cell segments (Fig. 1c) but also change the assignment of cell labels and ultrastructural features such as synapses (Fig. 1d) and require recalculations of morphological neuron features (for example, volume and area) and neuronal representations (for example, skeletons). Previous systems that combined reconstruction, annotation and analysis[32–35] only supported manual cell reconstructions and manual annotations.

As a second part of CAVE, we addressed the challenge of supporting analysis of proofreadable cell segmentations in conjunction with annotations produced by automated methods and individual users. CAVE enables fast computation of morphological neuron features

and representations at any time, including immediately after an edit, through an extension to the ChunkedGraph. We introduce a scheme for storing annotations, which binds annotations to segment IDs at specific points in time in a process we call 'materialization'. We show that CAVE's annotation and proofreading systems support fast queries of the data for any point in time by combining traditional database queries with ChunkedGraph-based tracking of neuron edit histories. This enables CAVE to answer analysis queries with no delays after an edit and queries of the dataset at arbitrary time points.

Together, CAVE manages concurrent proofreading, annotation and annotation assignment while offering queries to the data for any point in time to support flexible and reproducible analysis by a distributed group of users. CAVE is already used to host five published datasets where it tracks almost 2 billion annotations and has recorded over 4 million edits by over 500 unique users from across the globe. CAVE facilitated the reconstruction of the first whole-brain adult connectome with FlyWire[16,36] and supports the FANC community reconstructing the Drosophila VNC[37], proofreading in the H01 dataset[15] and the cubic millimeter-scale MICrONS volumes[14].

## Results

### Collaborative proofreading of petascale reconstructions

In a perfect segmentation, all voxels (three-dimensional (3D) pixels) within the same cell are labeled with the same ID (Fig. 2a). Here, we refer to a group of voxels with the same ID as a 'segment'. In an automated segmentation, most segments require subsequent proofreading to create accurate neuron reconstructions. Here, we use the term 'proofreading' exclusively for edits to the segmentation, but CAVE supports editing of annotations as well. Proofreading an automated segmentation is 10–100× faster than purely manual reconstruction[16,25,38], but proofreading of a large dataset may still go on for years. An ideal system should allow real-time collaboration by many proofreaders, including both humans and machines, and make the results available for concurrent analysis and discovery efforts.

These requirements are met by the ChunkedGraph proofreading system, whose design was described previously[28,29]. Like previous

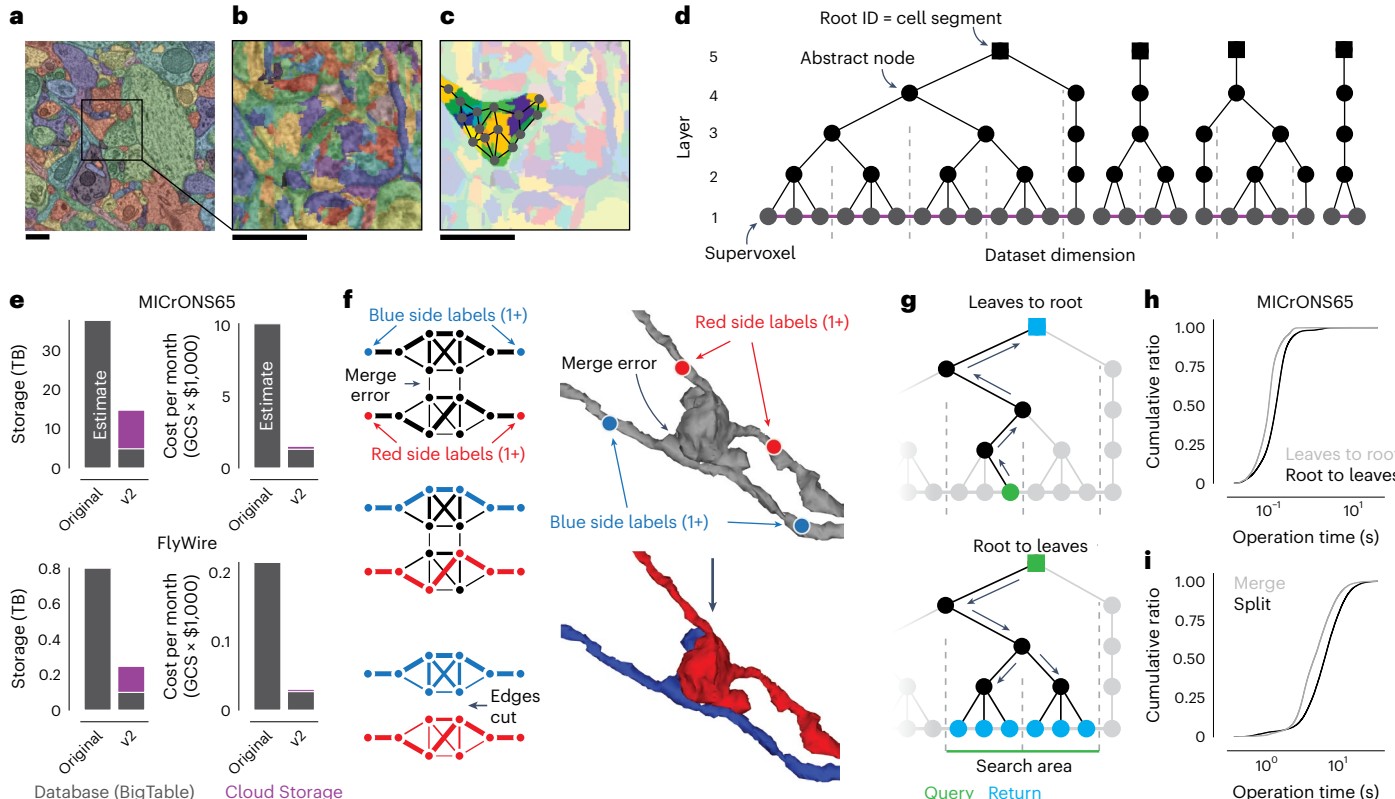

**Fig. 2 | Scaling the ChunkedGraph to petascale datasets. a**, Automated segmentation overlaid on EM data. Each color represents an individual putative cell. **b**, Different colors represent supervoxels that make up putative cells. **c**, Supervoxels belonging to a particular neuron, with an overlaid cartoon of its supervoxel graph. These data corresponds to the framed square in **a** and the full panel in **b**. **d**, One-dimensional representation of the supervoxel graph. The ChunkedGraph data structure adds an octree structure to the graph to store the connected component information. Each abstract node (black nodes in levels >1) represents the connected component in the spatially underlying graph. **e**, Storage and costs for the supervoxel graph storage under the original and the improved implementation (v2); GCS, Google cloud storage; TB, terabytes.

**f**, To submit a split operation, users place labels for each side of the split (top right). The backend system first connects each set of labels on each side by identifying supervoxels between them in the graph (left). The extended sets are used to identify the edges needed to be cut with a maximum-flow minimum-cut algorithm. **g**, Examples of graph traversals for looking up the root ID for a supervoxel ID (top) and supervoxel IDs for a root ID within a spatially defined search area (bottom). Note that only part of the graph needs to be traversed. **h,i**, Performance measurement from real-world user interactions measured on the ChunkedGraph server for different types of reads (**h**) and edits (**i**). The cumulative ratio of all measured interactions for a given response time is plotted on the *y* axis. Scale bar, 500 nm.

proofreading systems[26,30,31], the ChunkedGraph stores the segmentation as a graph of atomic segments, called supervoxels (Fig. 2a,b). Connected components in this graph represent cells (Fig. 2c). The ChunkedGraph introduced a representation of the segmentation as a spatially chunked hierarchical graph of supervoxels (Fig. 2d), where root nodes are individual cell segments, and leaf nodes are supervoxels. To achieve high performance, the ChunkedGraph requires a database featuring low-latency random row reads such as BigTable[39], which can add substantial cost to its deployment. CAVE uses the ChunkedGraph as proofreading backend and hosts it as a cloud service for world-wide access. Here, we describe two advancements to the ChunkedGraph to make it viable for petascale datasets.

First, we reimplemented the ChunkedGraph creating 'Chunked-Graph v2'. The initial ChunkedGraph version targeted proofreading of the FlyWire[29] and MICrONS phase 1 (https://www.microns-explorer.org/phase1)[13,28,40] datasets but turned out to be prohibitively costly for 50–100× larger petascale datasets like MICrONS65 (Fig. 2e). We redesigned the storage to a hybrid scheme in which supervoxel edges (Fig. 2d), which are only needed for edits, are compressed and stored on conventional storage while the octree hierarchy remains stored in BigTable (Fig. 2e; number of supervoxels in MICrONS65 = 112 billion). The reimplemented ChunkedGraph v2 reduced the cost by >6.5×, low enough to support proofreading at the petascale (Fig. 2e).

Second, we improved how user edits are processed to speed up proofreading. To make proofreading accessible, proofreaders should not need to be aware of the underlying data structures. Instead, users perform edits by placing line connectors for merges and points for splits (Fig. 2f). The ChunkedGraph implements splits with a maximum-flow minimum-cut operation where user-selected supervoxels are labeled as sources and sinks to find the edges in the graph that should be removed. Previously, many locations were needed to resolve complex false merges. To aid split operations, we implemented an algorithm that uses a small number of locations coarsely surrounding the merge error, making the resulting topology of the split robust to the precise location or number of labels (Fig. 2f and Extended Data Fig. 1). Split operations are always executed locally, and a user may need to execute multiple split operations to separate two cell segments that were falsely merged in multiple places. To speed up merging of many fragments, we added a multimerge operation to neuroglancer, allowing users to execute merge operations in parallel.

We measured view and edit performances during real-world proofreading of the ChunkedGraph v2 on the MICrONS65 dataset, the largest currently available, and the FlyWire dataset, which we could use for comparison with the original implementation. For viewing the segmentation, the user selects a supervoxel by clicking a location in space, and the system retrieves all supervoxels that belong to the same segment

within the field of view (Fig. 2g) in two steps. First, the ChunkedGraph is traversed from the selected supervoxel to the root node (Fig. 2g). For MICrONS65, the ChunkedGraph v2 responded with a median time of 69.5 ms and 95th percentile of 199 ms (server-side performance, $N = 91,976$; Fig. 2h). Second, the search proceeds down the hierarchy to retrieve all supervoxels within a bounding box around the user's field of view (Fig. 2g). Here, the ChunkedGraph leverages the octree structure to avoid the retrieval of supervoxels out of the user's view. We measured median response times of 105 ms and a 95th percentile of 290 ms (batched requests for multiple segments, $N = 184,821$; Fig. 2h). Next, we tested edit operations. The server completed merge operations with a median time of 4,116 ms ($N = 25,854$; Fig. 2i) and splits with a median edit completion time of 5,813 ms (server-side, including the logic to identify sources and sinks; $N = 21,920$; Fig. 2i). Repeating this analysis with the FlyWire dataset shows that viewing operations performed equally for the ChunkedGraph v2 and the original version, whereas edit operations showed a modest slow down of ~1.5 s (Extended Data Fig. 2). Notably, the performance for edits was only ~1.6× times slower on the MICrONS65 dataset, even though it is 67× larger than FlyWire, illustrating the scalability of the ChunkedGraph system.

## Morphological analysis of proofread neurons

Proofreading is often driven by specific analysis goals. Being able to analyze cells as they are being corrected is important for analysis and to guide further proofreading. For instance, morphological information about a cell (for example, volume and area as well as morphological representations such as skeletons and meshes) is used in many analyses. Skeletons are sparse representations of neurons that have proven useful for analysis and matching of neurons between datasets, including datasets of different modalities[41,42]. Computing these measurements and representations usually requires loading the entire segmentation of a cell, which can span a large part of a dataset. Recomputing these features from scratch after every edit is prohibitively time consuming and costly.

We leveraged the ChunkedGraph tree structure to cache and reuse meshes and morphological features for spatial chunks and only recompute features in the regions of a cell that changed due to an edit. This L2-Cache (Fig. 3a), named after its use of level 2 of the ChunkedGraph hierarchy, is populated automatically through a queuing system after every edit (Extended Data Fig. 3). Every edit produces a list of new level 2 nodes and associated level 2 IDs (L2 IDs), for which a scalable microservice computes new meshes and a set of features, for example, volume, area, representative coordinate and principal component analysis components (Extended Data Fig. 3).

Combined with a fast retrieval of all L2 IDs belonging to a neuron (Extended Data Fig. 4a), morphological features can be computed quickly. For instance, volume information can be computed within a median client-side time of 710 ms for FlyWire neurons and 3,176 ms for neurons in MICrONS65 (Fig. 3c). The longer times for MICrONS65 can be explained by the larger size of the neurons (Extended Data Fig. 4b).

To produce skeletons, the ChunkedGraph exposes a graph between L2 IDs, the L2-graph, which, when combined with locally computed representative coordinates from the L2-Cache, allows for rapid production of topologically correct skeletons (Fig. 3b). We implemented a graph-based generalization of the TEASAR skeletonization algorithm[43] on the L2-graph to remove short and artificial branches introduced by the L2 chunk boundaries. Skeleton calculations of neurons in MICrONS65 and FlyWire took a median of 5,996 ms and 1,171 ms (Fig. 3d), respectively, with differences again being explained by the difference in size (Fig. 3e and Extended Data Fig. 4c).

## Annotation schemes for rapid analysis queries

CAVE supports a diverse set of annotations from manual and automated sources. Every annotation is based on points in space (≥1) that serve as spatial anchors and are accompanied by a set of data entries (Fig. 4a).

Data entries range from biological measurements (for example, synapse size) to text annotations (for example, cell types). When creating an annotation table, a user selects a schema that defines which data columns need to be filled for each annotation (Extended Data Fig. 5a). CAVE has a repository of schemas from which users can freely choose, with many being reused across tables, datasets and communities. Users can create new schemas that fit their specific needs but are encouraged to reuse schemas where possible. To associate annotations with segments, the spatial points are bound to the underlying supervoxels, which can then be mapped to their associated root segment for any point in time using the ChunkedGraph (Fig. 4b–e and Extended Data Fig. 5b). For instance, schemas describing synapses between two neurons contain two 'bound spatial points', which are associated with the pre- and postsynaptic segments but vary in their additional parameters (for example, size and neurotransmitter; Extended Data Fig. 5c). We identified the need for allowing users to add data entries to existing annotations without needing to copy the entire table. For this, we implemented reference schemas that define reference tables. A reference table is linked to another table via foreign key constraints (SQL) and may only contain data entries for a subset of the annotations in the referenced table (Extended Data Fig. 6).

Compared to other tools[27], CAVE is designed to be flexible about how users define their annotations. This allows the system to capture an expanding set of rich observations about the dataset, from small ultrastructural details to observations about cell types and their anatomical locations. In fact, across MICrONS65 (ref. 14), MICrONS phase 1 (refs. 13,28,40), FlyWire[29] and FANC[37,44], users have created over 120 annotation tables (including 29 reference tables) using 21 distinct schemas and capturing over 1.8 billion annotations. This includes tables marking synapse detections, reference annotations on those detections, nucleus locations, 62 distinct cell-type tables, proofreading statuses, mitochondrial locations and functional co-registration points. A subset of these tables is associated with static volumetric segmentations of objects, such as mitochondria and synapses, which can be linked via the annotation ID to allow users to do volumetric analysis. We expect the diversity of observations to grow richer over time and as further secondary analyses are performed.

To manage annotations, CAVE maintains a 'live' SQL database of all annotations. Users create annotation tables with any schema to which they add, remove and update individual annotations. Every time an annotation is added or updated, supervoxels underlying any bound spatial points are automatically retrieved. The materialization service then frequently (for example, 1 per h) updates the associated root segments of all annotations using the ChunkedGraph (Fig. 4d,e and Extended Data Figs. 3 and 5b), thereby producing a materialized version of the annotation data.

## Queries for arbitrary time points

Data analyses require reproducible queries to the materialized annotation data. Because ongoing proofreading and annotating make the 'live' database unsuitable for analysis queries (Fig. 5a), we create infrequent copies (for example, daily; Fig. 5b) of it that serve as materialized analysis snapshots. Querying these snapshots ensures consistent queries but prohibits a user's ability to query the data immediately after fixing a segmentation error, a common scenario when doing exploratory analysis and proofreading. This makes the snapshot system unsuitable for managing proofreading, and a large number of snapshots would be required to support continued analysis of past time points.

To solve this problem, CAVE combines materialized snapshots with ChunkedGraph-based tracking of neuron edit histories to facilitate analysis queries for arbitrary time points (Fig. 5b). The ChunkedGraph tracks the edit lineage of neurons as they are being proofread (Fig. 5a), allowing us to map any segment used in a query to the closest available snapshot time point (Fig. 5a). This produces an overinclusive set of segments with which we query the snapshot database. Additionally,

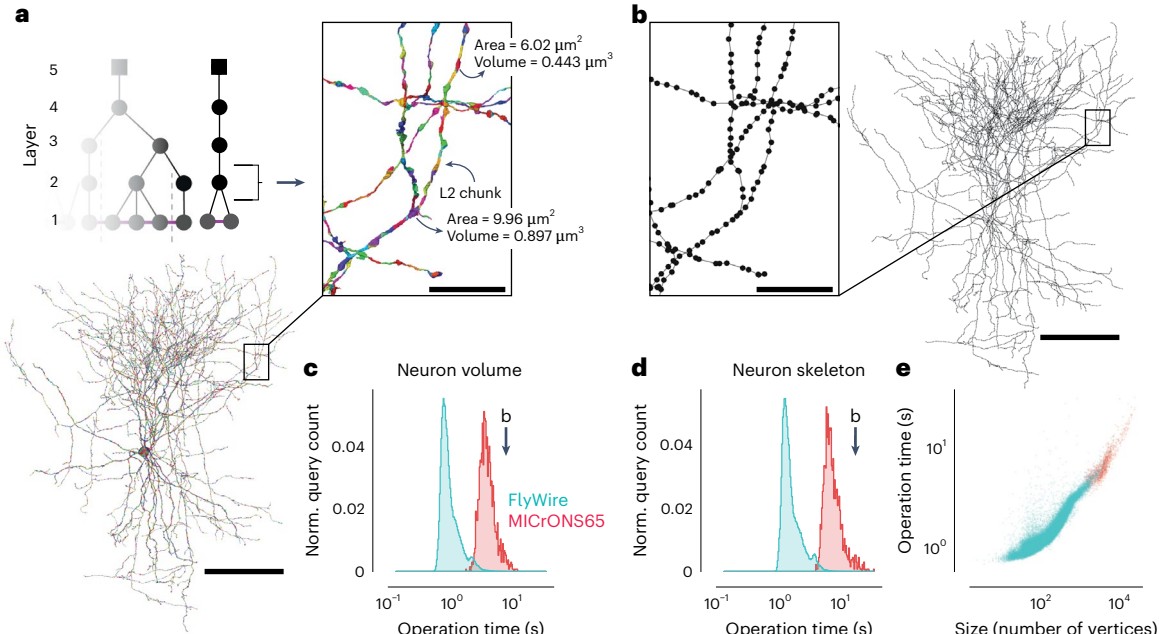

**Fig. 3 | Fast calculation of morphological features and skeletons. a**, The basket cell from Fig. 1c broken into L2 chunks where each chunk is colored differently. For each chunk, the L2-Cache stores a number of features, such as area, volume and representative coordinate. **b**, A skeleton derived from the ChunkedGraph and L2-Cache without consulting the segmentation data. **c**, Client-side timings for calculating neuron volumes using ChunkedGraph and L2-Cache for neurons in FlyWire and MICrONS65 ($N_{FlyWire} = 101,554$; $N_{MICrONS65} = 1,357$). The timing for the neuron in **b** is highlighted. **d**, Client-side timings for creating skeletons from ChunkedGraph and L2-Cache ($N_{FlyWire} = 78,030$; $N_{MICrONS65} = 1,357$). Norm., normalized. **e**, Client-side timings for creating skeletons plotted against the size of the skeletons. Each dot is a query for a single neuron (see **d** for the number of samples). Scale bars, 100 μm (insets: scale bar, 5 μm).

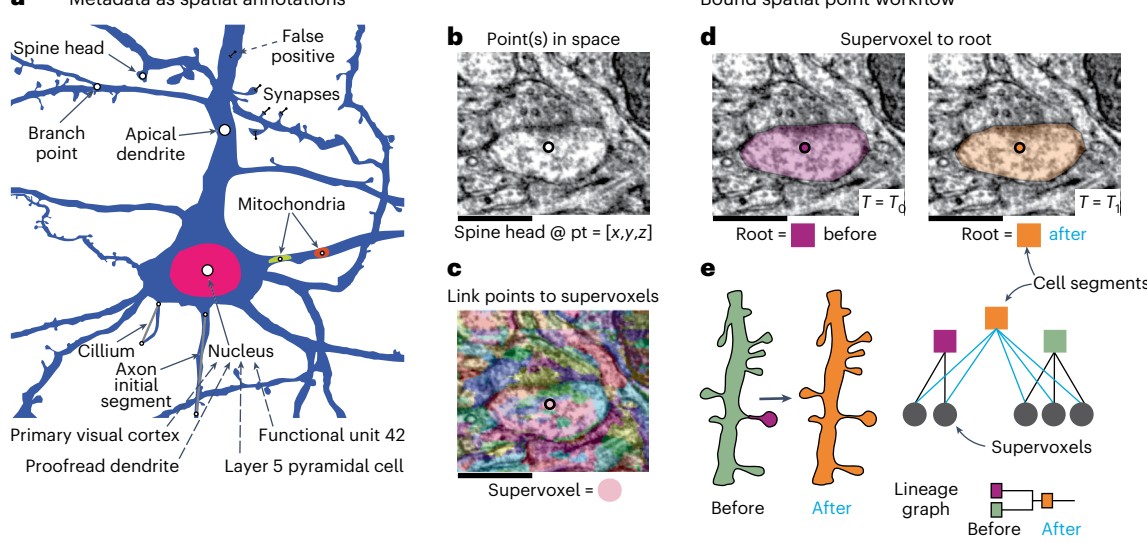

**Fig. 4 | Annotations for proofreadable datasets. a**, Spatial points can be used to capture a huge diversity of biological metadata generated by either human annotators or machine algorithms. Additional metadata for existing CAVE annotations can be added with reference annotations that avoid duplicating existing annotations (illustrated as dashed lines). **b**, The annotation services handle all annotations through a generic workflow that depends only on expressing all annotations as collections of spatial points and associated metadata. Spatial annotations mark the location of a feature, such as a spine head. Scale bar, 500 nm. **c**, The materialization service retrieves the supervoxel ID underlying all spatial points. **d**, This enables the materialization service to look up the root ID underneath that points at any given moment in time using the ChunkedGraph. **e,** Illustration of how the mapping from supervoxel ID to segment ID changed the annotation due to proofreading (octree levels not shown). The changes are tracked in a lineage graph of the altered roots.

we query the 'live' database for all changes to annotations since the used materialization snapshot and add them to the set of annotations. The resulting set of annotations is then mapped back to the query timestamp using the lineage graph and supervoxel to root lookups and finally reduced to only include the queried set of root IDs.

The additional logic required to execute arbitrary time point querying introduces an overhead over querying materialized analysis snapshots directly. To quantify this overhead, we turned to the FlyWire dataset for which numerous actively proofread neurons were available. Starting from a materialized snapshot, we queried presynapses (Fig. 5c)

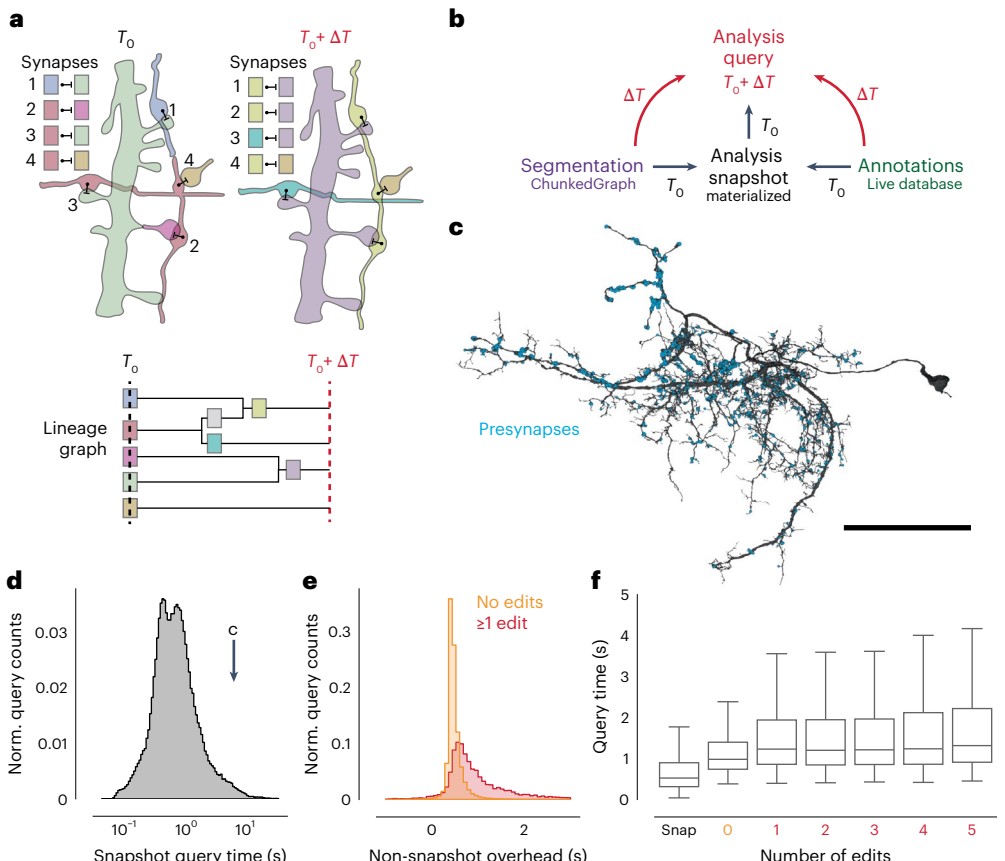

**Fig. 5 | Querying the dataset for any time point. a**, Edits change the assignment of synapses to segment IDs. Each of the four synapses is assigned to the segment IDs (colors) according to the presynaptic and postsynaptic points (point, bar). The identity of the segments changes through proofreading (time passed: $\Delta T$) indicated by different colors. The lineage graph shows the current segment ID (color) for each point in time. **b**, Analysis queries are not necessarily aligned to exported snapshots. Queries for other time points are supported by on-the-fly delta updates from both the annotations and segmentation through the use of the lineage graph. **c**, A neuron in FlyWire with all its automatically detected presynapses. **d**, Time measurements for snapshot aligned queries of presynapses for proofread neurons in FlyWire ($N = 121{,}400$). **e**, Difference between the snapshot and nonsnapshot aligned presynapse queries. The two distributions differentiate cases without any edits to the queried neurons and cases with at least one edit to the queried neuron ($N_{\text{no edits}} = 98{,}367$; $N_{\geq 1\,\text{edit}} = 8{,}132$). **f**, Presynapse query times for snapshot and nonsnapshot aligned queries, including cases where neurons were proofread with multiple edits. The horizontal bar is the median. Boxes are interquartile ranges, and whiskers are set at 1.5× the interquartile range. Number of samples by number of edits: snap, $n = 121{,}389$; 0, $n = 137{,}866$; 1, $n = 7{,}074$; 2, $n = 3{,}512$; 3, $n = 2{,}074$; 4, $n = 1{,}325$; 5, $n = 850$. Scale bar, 50 µm.

for individual neurons at several time offsets from the snapshot using the delta query logic. The resulting measurements can be categorized into three groups. First, we obtained measurements for presynapse queries of FlyWire neurons that were aligned with a snapshot (median = 525 ms, $N = 121{,}400$; Fig. 5d). Second, we gathered timings for nonsnapshot aligned queries where the query neuron did not see any edits since the snapshot version, although its synaptic partners may have (median = 978 ms, $N = 127{,}775$). Third, we gathered timings for nonsnapshot aligned queries where the query neurons were edited since the snapshot (median = 1,385 ms, $N = 12{,}303$).

By comparing measurements from the first two groups for queries to the same neuron, we can obtain the overhead of the additional logic (median = 447 ms; Fig. 5e). Although query times were well correlated with query size (Extended Data Fig. 7a), we found this offset to be largely constant across queries (Extended Data Fig. 7b). Queries are slowed down modestly for the third case where the queried segment changed since the last snapshot and an overinclusive query has to be generated (Fig. 5e,f).

## Modular and open design for broad dissemination

We designed CAVE along two broad principles: modularity and openness. Rather than a monolithic application, CAVE is designed as a set of loosely coupled services (Supplementary Table 1 and Extended Data Fig. 3). Each CAVE service serves a specific purpose, controls its own data and is deployed as a docker image to Google Cloud through kubernetes (Extended Data Fig. 8). Services can always be added to meet a specific need of a community and replaced with ones that fulfill the same purpose and application interfaces (APIs). CAVE builds on existing infrastructure and storage solutions as much as possible, from cluster management (kubernetes) to its databases (for example, redis and PostgreSQL). All services use off-the-shelf commercial and open-source components to manage their data. Although these are currently supplied through Google Cloud, all components can be replicated elsewhere, including on local servers.

CAVE services can be accessed through authenticated APIs. Permission levels can be adjusted for each user, endpoint and dataset with each user having no, view or edit access. We developed a Python client (CAVEclient) for programmatic access and adapted the popular viewer neuroglancer[45] (Fig. 6a) for interactive viewing and editing of the ChunkedGraph segmentation. Further, interactive analysis is enabled through custom dash apps (Python-based web apps) that can be extended to serve the needs of any community (Fig. 6a). CAVE's APIs can be accessed through other tools (Fig. 6a) as long as they authenticate with CAVE's centralized authentication and authorization server. The

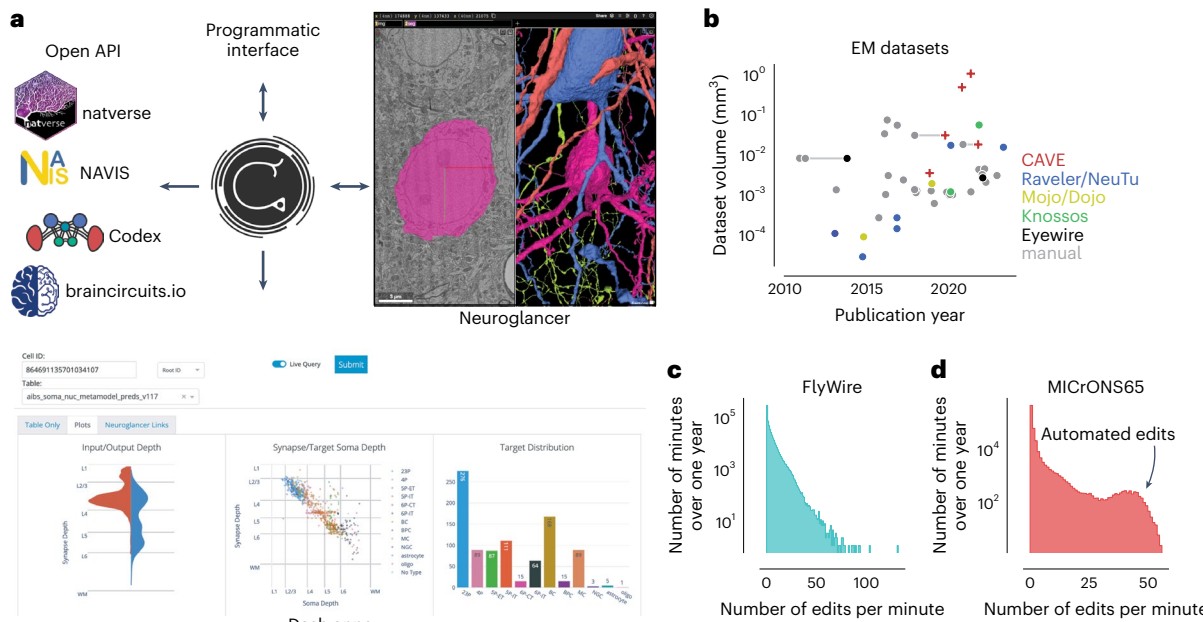

**Fig. 6 | Integration into connectomics projects. a**, CAVE supports multiple interfaces. In addition to programmatic access, users can explore and edit the data in CAVE interactively through neuroglancer or CAVE's dash apps. CAVE integrates with existing and new tools for connectomics through packages such as natverse[46], Codex and braincircuit.io. **b**, Datasets published since 2010 by volume and year (volume is plotted on a log scale). Datasets that were published with manual and semiautomated means are connected with a horizontal gray line (Supplementary Table 2). **c**,**d**, Proofreading rate in edits per minute for FlyWire (N = 1,349,955; **c**) and MICrONS65 over 1 year of proofreading (N = 457,285; **d**).

analysis package natverse[46] and the web applications Codex (https://codex.flywire.ai), braincircuits.io (https://braincircuits.io) and Neu-Vue[47] already serve as such examples (Fig. 6a).

To date, CAVE has facilitated proofreading and analysis of five published datasets with others in progress (Fig. 6b), including FlyWire[29], FANC[37], the MICrONS datasets[13,14,28,40] and the H01 human dataset[15]. Together, these communities accumulated over 4 million edits so far from over 500 unique users across the globe. Proofreading by a community puts unpredictable loads onto CAVE. Proofreading rates vary throughout the day, with FlyWire seeing as many as >100 edits per min (Fig. 6c). More than 150,000 edits in MICrONS65 were applied automatically[48] (Fig. 6d), illustrating how CAVE supports both manual and automated proofreading efforts.

## Discussion

We introduced CAVE, an open-source software infrastructure for managing proofreading, annotations and analysis by a distributed group of scientists. It is a system that enables concurrent proofreading and annotation querying at arbitrary time points for seamless analysis and the only system that has successfully demonstrated proofreading of petascale connectomics datasets.

Although CAVE demonstrates advances, it also combines many features inspired by prior tools for distributed connectome analysis (see Beyer et al.[49] for a review). CAVE was particularly influenced by CATMAID[32], which enables collaborative annotation, manual neuron tracing and analysis, and was used by the *Drosophila* larva community[50] and initial manual reconstruction of the full adult fly brain dataset[38]. Similarly, webknossos[34], Viking[35] and Knossos[33] (https://knossos.app) support collaborative manual tracing and annotating. For distributed proofreading, Eyewire[30] is the closest precedent, as it distributed block-based proofreading to a community through an interactive browser interface with remeshing capabilities after edits. NeuTu[26] demonstrated neuron-based proofreading at scale for a restricted group of people that proofread multiple *Drosophila* datasets, including the hemibrain[12].

Analysis of proofread data has so far relied on static exports after proofreading is completed. For instance, NeuPrint[27] provides analysis of data after it has been proofread in NeuTu. In theory, it can also use exports and materialized snapshots from CAVE. Although the reliance on static data is limiting for use during proofreading, such tools can provide more complex analyses, such as graph queries, through preprocessing of the synapse graph[51], as illustrated by NeuPrint[27] (https://neuprint.janelia.org), Codex (http://codex.flywire.ai) and FlyBrainLab[52].

Connectomics data, and biological imaging data in general, are being generated at a growing rate. Due to their size, these data are increasingly analyzed by multiple people for a long period of time, raising the demand for interoperable and flexible tools that enable simultaneous editing and distributed analysis across multiple user groups. In our observation, each community differs in the policies implemented that guide proofreading and analysis, raising the need for specific tools to organize a community around these policies. Here, CAVE's open API enables anyone to create light-weight services that interface with it, lowering the barrier to extend CAVE with analysis and community organization tools (see NeuVue[47] for one such example).

CAVE is generally applicable to any image dataset that requires the correction and annotation of automated segmentation by multiple users in parallel. Although EM connectomics has led the way in scaling to large volumes of highly resolved imagery, light microscopy methods using expansion of the tissue have emerged. These now enable dense, high-resolution imaging of small blocks of tissue[53] or sparse imaging of large volumes up to an entire mouse brain[54]. As reconstruction of these datasets progresses, infrastructures like CAVE will also become important for analysis.

In scaling up to petascale datasets, CAVE faced trade-offs between cost, operational complexity and performance. In particular, to deploy CAVE, a scientific project needs personnel that are able to manage container-based web services as opposed to standalone desktop tools. Furthermore, we optimized CAVE for large datasets with dense automated reconstructions and many users. This led us to focus our

engineering efforts on making CAVE scale well with respect to cost while maintaining sufficiently high performance, as illustrated by our upgrades to the ChunkedGraph v2.

Scaling CAVE to future ten times larger datasets will require further upgrades to the ChunkedGraph and the way annotations are stored in our SQL databases. Scaled naively from MICrONS65, such a dataset would contain ~10 trillion supervoxel edges, which require large amounts of storage even with the improvements presented here. Larger supervoxels would reduce storage costs but inevitably lead to some supervoxels containing a merge error and needing to be split to accurately correct a segmentation. Therefore, the ChunkedGraph will need to implement supervoxel splitting to allow supervoxels to be grown in size and reduce their number. Another substantial source for cost is the storage of all annotations in a relational database that can be quickly queried. Automated pipelines now provide accurate and valuable annotations at scale[1,55], but storage costs grow linearly with the number of annotations and snapshots. With the emergence of multidimensional annotations demonstrating efficient prediction of semantic information[56], storage solutions, such as Occasionally Cooperative Distributed B-Trees (https://github.com/google/tensor-store), will be needed to leverage their power while keeping storage cost in check.

Despite the speed up provided by proofreading of automated segmentations over manual tracing, the manual proofreading component of the reconstruction pipeline remains one of the costliest and slowest steps in the dataset creation process. Further advancements in automated reconstruction will be needed to enable scaling to larger datasets[57]. Instead of improving the automated segmentation directly, tools to automate the proofreading process are emerging[48,58]. Their application will require new workflows of human–artifical intelligence interaction[47]. Here, CAVE's services have already served as a backend system to ingest edits from one such automated proofreading pipeline[48]. Due to its broad use, CAVE systems are already holding on to a wealth of data, both annotations and edit histories, that should be leveraged by automated methods to predict rich annotations of the data and help reduce the need for manual proofreading.

## Online content

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

¹Princeton Neuroscience Institute, Princeton University, Princeton, NJ, USA. ²Computer Science Department, Princeton University, Princeton, NJ, USA. ³Allen Institute for Brain Science, Seattle, WA, USA. ⁴Google Research, Mountain View, CA, USA. ⁵School of Engineering and Applied Sciences, Harvard University, Boston, MA, USA. ⁶Department of Biology, Lund Vision Group, Lund University, Lund, Sweden. ⁷Research & Exploratory Development Department, Johns Hopkins University Applied Physics Laboratory, Laurel, MD, USA. ⁸Electrical and Computer Engineering Department, Princeton University, Princeton, NJ, USA. ⁹Brain & Cognitive Sciences Department, Massachusetts Institute of Technology, Cambridge, MA, USA. ¹⁰These authors contributed equally: Sven Dorkenwald, Casey M. Schneider-Mizell. ✉e-mail: forrestc@alleninstitute.org

## Methods

### Authentication and authorization

The middle-auth service provides a dataset-specific authorization layer on top of OAuth2-based authentication. End points allow services to query whether users have different permissions on different service tables. The middle-auth service provides a mapping between service tables and datasets as well as individual users and groups. Groups then have permissions on datasets.

For example, the ChunkedGraph service has a table named minniev1, so when a user attempts to perform an edit on that table, the service will query a middle-auth end point to inquire if that user has 'edit' permissions on that table. First, if the user is not logged in, middle-auth will forward them onto Google's OAuth2 service to authenticate their identity. After return, that user is then registered with a unique ID in the middle-auth system. The 'minniev1' string is mapped in the 'microns' dataset in the ChunkedGraph service namespace, and all the permissions that groups the user is a member of are gathered to see if at least one of them has edit access. If it does, the middle-auth end point returns a success; otherwise, it returns an unauthorized status code, which is forwarded to the user. This same workflow is used whether or not the user is interacting with the service via Python or via neuroglancer.

The programming of this interaction is simplified by the middle-auth client library, which provides a set of decorators that can be used on flask end points to ensure that users are logged in or that they have particular permissions enabled to access that end point. The user's ID is then made available in the flask global variable dictionary for the service to record which user is performing each request.

### Microservice architecture

Services are run in docker using a nginx-uwsgi implementation to distribute requests to multiple worker processes operating in a single container. Generally, services have been written in Python using the Flask framework, with varying Flask plugins used by different services. We use kubernetes to manage container deployment. Kubernetes spins up multiple container pods to increase the number of requests that are handled by each service. Requests are distributed across those pods through load balancing, and an nginx-ingress controller is used to route requests from a single IP to the appropriate service based on the URL prefix. Most CAVE services are implemented with a common set of technologies and patterns, although this is not strictly a technical requirement. Cert-manager is used in conjunction with CloudDNS to manage and renew SSL certificates.

### Neuroglancer state server

The neuroglancer state server was written as a Python flask app, where the json states are stored in Google Cloud Datastore as a simple key-value store. Keys are state IDs, and values are the json encoded neuroglancer state. By passing states through an end point, programmatic migration of old state values and formats was possible and enabled seamless changing of user experiences as systems migrated. Posting and retrieving json states are implemented as separate end points. In neuroglancer, a 'Share' button uses the posting end point to upload the json state, which returns a state ID. The user is provided with a shortened link that contains a reference to the retrieval end point with that state ID as a query argument. We programmed neuroglancer to automatically load states when they are defined in the URL, and so this mechanism effectively allows the reduced size link to be shared easily even when the number of annotations specified in the state is very large.

### ChunkedGraph implementation

The original ChunkedGraph implementation is described elsewhere[29], and all concepts described there also apply to the ChunkedGraph v2. The ChunkedGraph implements the graphene format, which is derived from neuroglancer's precomputed format used for common segmentations.

**Supervoxel edge storage and retrieval.** In the original implementation, all supervoxel edges were stored in BigTable. We devised a storage scheme where all edges are stored on GCS. Edges are only accessed for edits. Edges are sorted by chunk and stored as protobufs of compressed arrays. Arrays are compressed using zstandard[59]. We changed the edge reading logic to read edges from a chunk in bulk to minimize access to GCS.

Cross-chunk edges are accessed more often than edges within chunks because they link individual components from subtrees together. They are also used to create an L2-graph, a graph between all L2 chunk components. Because of that, cross-chunk edges are also stored in BigTable.

Edges are either 'on' or 'off', and only the 'on' edges contribute to the connected components. The initial ChunkedGraph implementation stored this information alongside the edges. This information was redundant with the ChunkedGraph's hierarchy. In ChunkedGraph v2, we do not store 'on'/'off' information with the edges. We implemented logic to infer the edge state from the ChunkedGraph's hierarchy directly.

The edge information on GCS is never changed. If new edges need to be inserted, they are stored in BigTable. We call such edges 'fake edges' and reserve one row per chunk for them. Every operation reading supervoxel edges from GCS also checks for fake edges. Like all entries in BigTable, fake edges are timestamped, allowing for an accurate retrieval of the supervoxel graph for timestamps before their creation.

We implemented new logic to check whether fake edges are needed. When a user commands a merge operation, we first check if there is a path in the local supervoxel graph. If there is, we extract all local edges between the two components that are being merged and process the merge operation with them. If there are no such edges, the original user input is used to insert a fake edge between the two selected supervoxels.

**Common format for all supervoxel graphs.** The reimplemented storage of the supervoxel edges does not change and can be used for multiple ChunkedGraphs for the same dataset. This further reduces the per-ChunkedGraph cost. This edge storage also serves as a common format from which the ChunkedGraph can ingest supervoxel graphs created by other segmentation pipelines. In addition to edges, this format also contains component files (protobuf), which store the connected component information inferred by the segmentation pipeline. So far, exports to this format were generated for the flood filling segmentation pipeline[22] and the LSD segmentation pipeline[23].

### ChunkedGraph Meshing

**Meshing procedure.** We use zmesh (https://github.com/seung-lab/zmesh) for meshing of the segmentation. Every L2 chunk component is meshed at MIP 2, and the mesh is stored on GCS. Some L2 components only consist of a few pixels and might not be meshed at all. L2 meshes are then stitched on the layers above following the ChunkedGraph's hierarchy up to a stop layer (for example, 6), at which many meshes are too large to be held in the memory of a worker node or time constraints of queueing systems (for example, Amazon SQS) are surpassed.

**Mesh storage.** The large number of L2 IDs ($>10^9$) translates into a large number of L2 meshes, which would be expensive to store as individual files on GCS because of the high cost of write operations (for example, GCS charges $0.005 per 1,000 write operations). For the initial meshing of all segments in the ChunkedGraph after ingest, we store meshes in sharded files. For each L2 chunk, we store all meshes in one sharded file using cloudfiles (https://github.com/seung-lab/cloud-files). This pattern is repeated for higher layers, where stitched versions of meshes are stored, reducing the number of files that need to be downloaded for any given neuron. The sharded format allows retrieval of byte ranges from each shard but adds two additional reads to the header for retrieving

the byte range. Note, this is a different arrangement from the original precomputed sharded mesh format, where all the mesh fragments from a single neuron can be found in a shard.

Sharded files cannot be extended, only rewritten. This posed a problem for meshes that were created for new components after an edit. However, these are few in number compared to the number of meshes from the initial meshing run. Consequently, we store each of these newly generated mesh fragments as a single file on GCS. We refer to this format as 'hybrid mesh format' because it uses both sharded and single-file storage.

Each mesh is compressed using the Draco format for which we wrote and maintain a Python client (https://github.com/seung-lab/DracoPy). The Draco format is a lossy mesh compression format where every mesh vertex is moved to the closest grid node. The grid's spacing determines the compression factor. We place a global grid onto the dataset such that meshes retrieved from different chunks can be merged through overlapping vertices.

**Mesh retrieval.** Neuroglancer's precomputed format requires a manifest per segment outlining the mesh fragments that need to be read from GCS to produce a complete rendering of a segment. Every edit creates a new cell segment ID with a new manifest. Instead of precalculating and storing all manifests, the ChunkedGraph produces manifests for segments on the fly from the hierarchy of a neuron. Using the timestamp of the fragment ID, the ChunkedGraph can determine whether the fragment is stored in sharded or file-based storage and provide instructions accordingly.

Cloudvolume implements all necessary interactions with the ChunkedGraph and can be used to programmatically read meshes. MeshParty wraps this functionality and adds convenience functionality, such as caching of meshes, on-disk and in-memory, and provides further capabilities, such as mesh rendering using VTK.

## L2-Cache

Features for each L2 ID are calculated on the binarized segmentation. For each L2 ID, we currently calculate the following features: representative coordinate, volume, area, principal components, mean and maximum value of the euclidean distance transform and number of voxels at each chunk boundary intersection.

Area calculations are difficult to perform and are easily inflated by rough surfaces. However, smoothed measurements are ill defined and expensive to obtain. Thus, our area measurements overestimate the actual area of a neuron. We calculate areas by shifting the segmentation in each dimension and finding all voxels where the segment of interest overlaps with other segments. We count up the surfaces and adjust for resolution. L2 features are stored in BigTable. Every L2 ID matches to a row in BigTable and contains a column for each feature.

## Edit-based triggers for data generation

Edits require the calculation of new meshes and L2 features for all newly generated L2 IDs. This is an expandable set of work tasks that we keep independent from the ChunkedGraph service to increase modularity and expandability. After every edit, the ChunkedGraph adds a message to a queuing system (Extended Data Fig. 3) to which individual services can subscribe. We use Google's Pub/Sub.

## L2 skeletonization

Skeletons were generated using a graph-based generalization of the TEASAR algorithm[43] using L2 chunks. For a given root ID, we query the ChunkedGraph for its component L2 IDs and the list of which L2 chunks are directly adjacent to others, either via supervoxels that spanned chunk boundaries or proofreading edits that introduced edges between chunks. We next query the L2-Cache to identify the representative coordinate (and other properties) for each L2 ID and use this information to generate a graph where each vertex is a single

L2 chunk and edges have a weight given by the distance between representative coordinates of adjacent chunks. Following the TEASAR algorithm, we identify a root node (for example, the closest vertex to a cell body centroid or the base of the axon from a peripheral sensory neuron) and find the most distant vertex on the graph. The vertices along the shortest path from the distant vertex to the root node are assigned to the skeleton, and we 'invalidate' vertices within a distance parameter provided by the user. Importantly, we store a mapping from each invalidated graph vertex to the closest skeleton vertex. We iterate this process using the most distant uninvalidated vertex and the shortest path to the existing skeleton until all vertices are invalidated.

To associate synapses with vertices on the skeleton, we get the supervoxel ID of the bound spatial point(s) associated with the annotation and use the ChunkedGraph to look up its associated L2 ID(s). We then assign synapses to graph vertices via L2 ID and use the invalidation mapping to associate graph vertices with supervoxel vertices. Similarly, L2 properties, such as volume or surface area for regions of a skeleton, can be computed by summing the appropriate values from the L2-Cache via the associated graph vertices. The core skeletonization process was implemented in MeshParty, and the interaction with the ChunkedGraph is handled through the Python library 'pcg_skel'.

## Schema implementation

Annotation schemas are defined in Python code, where they are constructed using the Marshmallow library. Each schema contains at least one field of the custom class 'BoundSpatialPoint'. This field implicitly creates fields for positions, supervoxel and their associated root IDs. The annotation and materialization process can thus also dynamically locate BoundSpatialPoint fields and use them to execute the generic workflow of supervoxel and root ID lookup described in the materialization process.

Reference annotations were defined as a custom schema subclass with a target ID field associated with them. Postgres data access and storage was facilitated by a module that automatically constructed SQLalchemy models, using GeoAlchemy to describe spatial positions as spatially indexed 3D points. This model creation code automatically adds spatial indices to spatial points and SQL indices to associated root ID columns to facilitate fast querying. It also generates the foreign key constraints associated with reference annotations. Each schema is assigned a unique string for identification and is used by libraries to indicate what schema a table uses. Reference annotations may or may not have their own set of BoundSpatialPoints, and their model creation requires an extra parameter to create a foreign key between the target ID column of the reference table and the ID column of the table that is being referenced.

The EMAnnotationSchemas repository is the source of truth for what kinds of schemas can be initialized, and the community can contribute suggestions through pull requests to this library. Because all model creation code is written generically, extending the schemas supported is easy. This code is then used both as a library in other services and as a flask-based web service that makes a dynamic list of schemas and their structure as jsonschema, facilitated by the marshmallow-jsonschema library available.

## Annotation service

The annotation service manages the creation of new annotation tables and creation, deletion and updating of annotations within tables (Extended Data Figs. 3 and 6). Annotation tables are stored in Google Cloud SQL using PostgreSQL through a library called DynamicAnnotationDB. Annotation tables of any schema can be created, and multiple tables with the same schema may exist. When creating a table, users provide metadata about the table via a REST end point. These include a description, read and write permissions and the resolution of the spatial points. The permission model currently allows for three levels of permission for both read and write. 'PRIVATE' allows only that

user to read or write, 'GROUP' allows for users in the same group (see authorization) to read or write, and 'PUBLIC' allows for all users with read or write permissions on the dataset to read or write that table. The default permissions are 'PRIVATE' write but 'PUBLIC' read to encourage data sharing and reuse within communities.

If the user is creating a table with a reference schema, then they also must specify the name of the table that is being referenced. The service then uses the DynamicAnnotationDB library to create the table within the live SQL database and stores the metadata about the table in a separate metadata table. Annotations can be posted through a separate end point, which accepts json serialized versions of annotations. Annotations are then validated against the schema using marshmallow, and the SQLalchemy model is dynamically generated by the schema library. Annotations are then inserted into the PostgreSQL database after associating a creation timestamp to the annotation.

After insertion, the annotation service sends a notification to the Materialization service to trigger supervoxel lookups for the recently added annotations. Deletion is implemented virtually by marking the timestamp of deletion to enable point-in-time consistent querying. Updates are represented as a combination of remove and add operations. The CAVEclient has Python functions for facilitating client-side validation and packaging of annotations for the REST end point, including support for processing pandas dataframes. In addition to the API, the service provides a human readable website interface for browsing existing tables.

## Materialization implementation and annotation databases

The Materialization Engine updates segmentation data and creates databases that combine spatial annotation points and segmentation information. There are two types of databases that the system uses, one database for the 'live' dataset and multiple for materialized snapshots (Extended Data Fig. 3). The live database is the one written to by the Annotation service and is actively managed by the Materialization service to keep root IDs up to date for all BoundSpatialPoints in all tables (Extended Data Fig. 5). The data written by the Materialization service (segmentation data are in Extended Data Fig. 5) are stored in a separate table to separate concerns and to allow the same annotation data to be used by different Materialization services. Snapshotted databases are copies of a time-locked state of the 'live' database's segmentation and annotation information used to facilitate consistent querying.

To keep the data in sync, the backend leverages Celery, a Python-based distributed task queue, which allows for scaling and distributing parallel workloads. Using dynamic Celery-based workflows, the Materialization Engine runs periodic tasks to keep the segmentation information up to date from the proofreading efforts and provides copies of snapshotted databases at a fixed interval for analysis.

The Materialization Engine is deployed to a kubernetes cluster where Celery is run on pods. Two types of Celery pods are deployed for CAVE: producers and consumers. Producers create workflows that dynamically generate tasks that the consumer pods will subsequently execute.

## Annotation query implementation

The Materialization service also provides a query API for users to query both materialized versions of the database as well as an end point that implements a workflow that enables arbitrary moment-in-time querying of the data, with nearly identical features. Both sets of end points enable arbitrary filters on columns from the annotation tables, with inclusive, exclusive and strictly equal filter options as well as bounding box spatial queries on all spatial points. By filtering on the segment ID columns of associated BoundSpatialPoints, users can efficiently extract all annotations for individual cells. For example, this lets users retrieve all input or output synapses from a particular set of neurons or allows users to query what cell-type annotations are associated with a particular set of cells. A join query end point allows

users to create arbitrary inner join queries on annotation tables with the same filter criteria. Queries return data either as PyArrow binary dataframes, which are faster and more efficient, or json serialized objects, which are more cross-platform compatible depending on a query parameter option. To prevent queries from accidentally requesting multiple gigabytes of data, an arbitrary configurable upper bound on the number of rows that are requested from the SQL database is enforced. Presently, our deployed systems have configured this to be 500,000 rows, although users can distribute more data by executing multiple requests in parallel.

Although the live query end point appears similar to the materialized end point to the user, the workflow in the background is more complex. In addition to the filters described above, users must specify a timestamp that they are interested in querying for a live query. First, the system uses the ChunkedGraph's lineage graph to translate all the filter parameters referencing segment IDs into an overinclusive set of related segment IDs that are present in the closest materialized time point. Equality filters are translated to inclusive filters in this process. This translated query is then executed against the materialized database to retrieve all the annotations that are potentially related to the user's query. For all the segment IDs that are not valid at the user-provided timestamp, the ChunkedGraph API is then queried using the associated supervoxel IDs to determine the correct segment ID for those BoundSpatialPoints. This covers any changes that might have happened in the segmentation data between materialization and the queried time point but does not account for changes in the annotation data that might have happened in that interval. Therefore, a second query is executed on the 'live' database using a filter on the created and deleted columns to extract any annotation rows that were added or removed on the queried tables. Note, if the closest materialization point is in fact in the future, then the meaning of addition and removal is inverted with respect to this step. The annotation service also tracks a timestamp for when a table was last modified to skip this step if there is no possibility that the table was altered in the interval between materialization and the query. Filters on segment IDs must be ignored in this process because there can be no guarantee of consistency on the live database due to ongoing and distributed update operations. Once these new and deleted annotation rows are retrieved, the same process of updating expired segment IDs using the ChunkedGraph API is applied. Rows from the materialized query that exist as deletions in the live database query are removed, and rows that were added are concatenated to the result. Finally, the original query filters on segment IDs are applied to these aggregated results to remove any annotations that are not relevant to the user's query.

## CAVE deployment and cost

Current deployments of CAVE are wholly based on Google Cloud, but most parts of CAVE are general such that they can be deployed with other cloud providers or locally. Costs per deployment scale with the following three main factors: (1) size of the supervoxel graph, (2) number of annotations and (3) number of users.

(1) The size of the supervoxel graph determines the amount of data stored in BigTable and regular storage (for example, GCS). Figure 2e provides cost numbers for the FlyWire and the MICrONS65 dataset. There is a lower cost bound to create a BigTable, and smaller datasets are typically below the threshold where more than one BigTable node is required. Multiple datasets can leverage the same BigTable instance, amortizing its cost across them. The numbers in Fig. 2e provide the cost for such a scenario.

(2) The number of annotations and the number (and type) of columns determine the storage requirements for the annotation databases. Scaling with the size of the dataset volume varies between datasets especially because synapse densities are

different by almost an order of magnitude between mammalian and insect brains. Users of CAVE can influence cost by setting the number of materialized databases that are being kept. MICrONS65 and FlyWire use databases with ~2 TB of storage, costing ~$500 per month. Smaller datasets will incur relatively less cost, and Google Cloud provides an easy-to-use calculator to determine a prospective cost.

(3) CAVE's microservices scale with usage and use tuned node pools (for example, preemptible, low memory and so on) to reduce cost. However, there is a lower cost bound because a deployment cannot scale below a single instance of each service. The lower cost of the services of a CAVE deployment is around $360 per month.

We provide extensive documentation for setting up and managing CAVE deployments. In our experience, an external lab requires a software developer or otherwise experienced technical person with programming and computer system knowledge. Under these circumstances, external labs were able to learn and set up CAVE deployments within a few weeks. To be explicit, the CAVE deployments managed by the CAVE team are not intended to host datasets from outside groups.

#### ChunkedGraph performance measurements
We measured server response times for all end points served by the ChunkedGraph from all users for several weeks while proofreading was progressing as normal. These numbers reflect real interactions and are affected by server and database load and are therefore an underestimate of the capability of our system. The most common requests are root to leaf requests as they are executed every time a user moves their field of view in neuroglancer. We sampled a random set of these interactions. For all others, we sampled all interactions.

#### Morphological feature performance measurements
We used a compute node on Google Cloud to execute programmatic queries to CAVE. First, we selected representative sets of neurons from each dataset (FlyWire and MICrONS65). We used neurons from MICrONS65 that were included in a recent circuit analysis[60], representing most proofread neurons in the dataset and all neurons that were marked as proofread from FlyWire. We then randomly sampled neurons from each list and queried (1) all L2 IDs from the ChunkedGraph and (2) all volume measurements from the L2-Cache for these L2 IDs. We finally added up all volume measurements for a total volume. We processed neurons sequentially for multiple days and recorded all measurements. We averaged time measurements for neurons for which we gathered multiple measurements. Measured performances were affected by the current load on the system.

#### Annotation query performance measurements
We used the proofread neurons from FlyWire for this analysis. Starting from materialization 571, we executed presynapse queries at time offsets of 0, 1, 10, 20, 40, 100, 400 and 800 h, recreating realistic queries. It should be noted that most queries are within 24 h of a materialization version. We precomputed the neuron IDs and number of edits after the materialization version for those time points and created a list of tuples containing (segment ID, timestamp), from which we randomly sampled entries and executed presynapse queries. Queries were executed sequentially.

#### Reporting summary
Further information on research design is available in the Nature Portfolio Reporting Summary linked to this article.

#### Data availability
The datasets used in this paper are publicly available. The MICrONS dataset[14] can be accessed at https://microns-explorer.org/cortical-mm3 and https://bossdb.org/project/microns-minnie. The FlyWire dataset[16] can be accessed at https://flywire.ai.

#### Code availability
All code is made publicly available via Git repositories using open-source licenses (MIT, MPL-2.0, GPLv3) at https://github.com/orgs/CAVEconnectome/repositories. We provide individual links to all repositories in Supplementary Table 2 and a description of all parts of CAVE at https://github.com/caveconnectome. Further, all docker images for CAVE services are publicly available at https://hub.docker.com/repositories/caveconnectome. Data and code to transform Source Data files into figures from the paper can be found at https://github.com/CAVEconnectome/CAVEpaper/.

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

#### Acknowledgements
We thank J. Wiggins, G. McGrath and D. Barlieb for computer system administration and M. Husseini for project administration. We thank S. Davis for the design of the CAVE logo. We thank G. Jefferis, D. Bock, E. Perlman, P. Schlegel and S. Gerhard for providing feedback on the system design. We thank G. Jeffries, P. Schlegel, B. Wester, S. Gerhard, A. Matsliah, B. Celli and J. Reimer for building tools that leverage the CAVE infrastructure and providing consistent feedback on its performance and implementation. We thank P. Nunez Gomez for providing advice about deployment strategies. We thank J. Tuthill (University of Washington), W.-C. A. Lee (Harvard Medical School) and the FANC community for their collaboration. We thank V. Jain (Google) and J. Lichtman (Harvard) for collaboration on the H01 dataset. We thank S. Heinze and K. Teodore (Lund University) for collaboration on their datasets. We thank Zetta AI for collaborations that use CAVE for further datasets. We thank the FlyWire Consortium for collaboration on the FlyWire dataset. We thank the Allen Institute for Brain Science founder P. G. Allen for his vision, encouragement and support. H.S.S. acknowledges support from the National Institutes of Health BRAIN Initiative RF1 MH129268, U24 NS126935 and RF1 MH123400 and assistance from Google. F.C., N.M.d.C. and R.C.R. acknowledge support from National Institutes of Health RF1MH125932 and National Science Foundation NeuroNex 2 award 2014862. J.T. and H.P. were supported by National Science Foundation grant NCS-FO-2124179. This work was supported by the Intelligence Advanced Research Projects Activity via Department of Interior/Interior Business Center contract numbers D16PC00004, D16PC0005, 2017-17032700004-005 and 2020-20081800401-023. The US Government is authorized to reproduce and distribute reprints for governmental purposes notwithstanding any copyright annotation thereon. The views and conclusions contained herein are those of the authors and should not be interpreted as necessarily representing the official policies or endorsements, either expressed or implied, of Intelligence Advanced Research Projects Activity, ODNI, Department of Interior/Interior Business Center or the US Government.

#### Author contributions
S.D., F.C. and C.M.S.-M. designed CAVE's core functionalities, service interactions and layout. S.D. and A.H. implemented the ChunkedGraph and the L2-Cache. M.A.C., S.D., W.S. and A.H. implemented the ChunkedGraph meshing logic. C.M.S.-M. implemented the improved splitting logic. C.J., N.K., J.M.-S. and D.X. extended neuroglancer

for proofreading. J.M.-S., V.G., J.T. and H.P. implemented adapters for and tested CAVE with supervoxel graphs produced by other segmentation pipelines. C.J., S.D. and F.C. implemented the authentication system. F.C. and D.B. implemented the annotation service and the annotation schema system. D.B., F.C. and S.D. implemented the Annotation database and the materialization service. S.D. and F.C. implemented the neuroglancer state server. F.C., C.M.S.-M., S.D. and D.B. implemented the CAVEclient. F.C., C.M.S.-M. and S.D. implemented MeshParty. C.M.S.-M. and F.C. implemented NeuroglancerAnnotationUI. A.H. and S.D. implemented datastoreflex. C.M.S.-M. and F.C. implemented PCGskel and skeletonization processing. C.M.S.-M. and F.C. implemented the dash apps. W.S. provided support and tools for fast cloud storage access. F.C., S.D., C.M.S.-M., D.B., C.J. and A.H. maintained the kubernetes deployments. A.H., A.L.B., B.N., C.J., C.M.S.-M., D.B., D.J.B., D.K., E.M., F.C., G.M., H.S.S., J.A.B., J.B., J.W., K. Lee, K. Li, L.E., M.A.C., M.T., N.K., N.L.T., N.M.d.C., R.C.R., R.L., R.T., S.-c.Y., S.D., S.K., S.M., S.P., S.S.M., T.M., W.S., W.W., W.Y. and Z.J. created the structural MICrONS65 dataset. S.D., F.C. and C.M.S.-M. wrote the paper with contributions from all authors.

## Competing interests

T.M., K. Lee, S.P., N.K. and H.S.S. declare financial interests in Zetta AI. S.D. and J.M.-S. are employees of Google, which sells cloud computing services. H.S.S. declares in kind donations by Google received as access to cloud compute resources. The other authors declare no competing interests.

## Additional information

**Extended data** is available for this paper at https://doi.org/10.1038/s41592-024-02426-z.

**Correspondence and requests for materials** should be addressed to Forrest Collman.

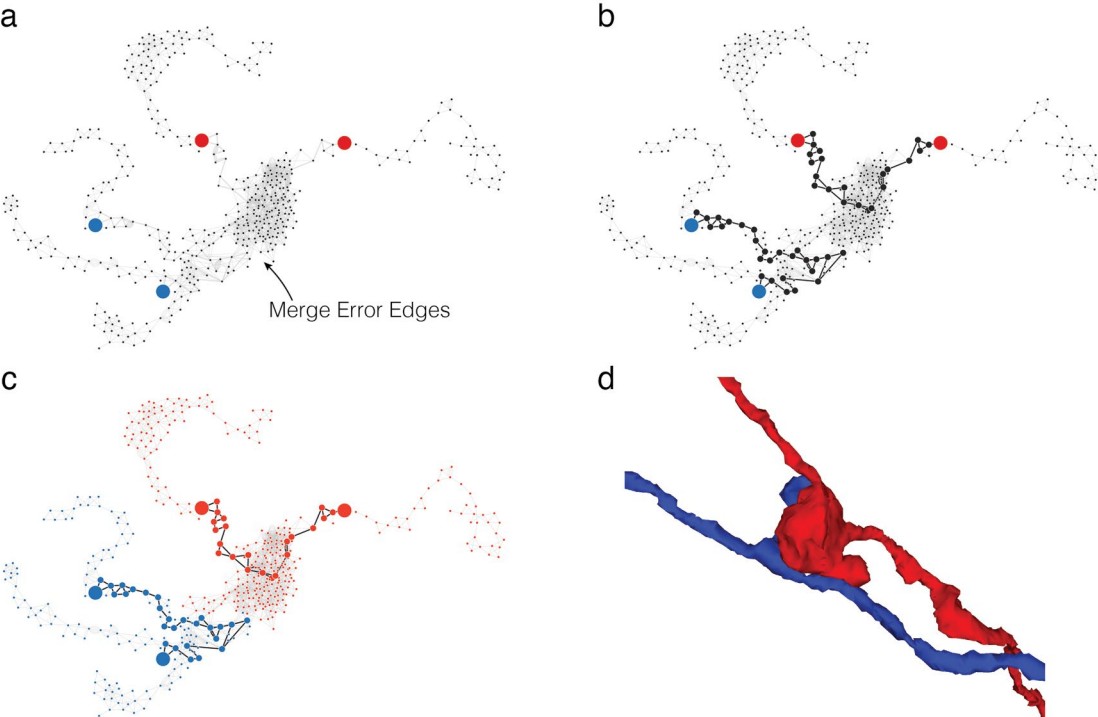

**Extended Data Fig. 1 | Translating user inputs to graph splits. (a)** Bipartite split labels are applied to locations in space. **(b)** The closest supervoxels to label points are identified (red/blue dots). The supervoxel graph in the neighborhood of the labeled points is computed (graph), weighted by affinity between supervoxels.

**(c)** Vertices along the shortest paths between each pair of red/blue labels are found (black dots and edges). Backup methods prevent overlap between paths. **(d)** Affinity between vertices along shortest paths is set to infinity and min cut is computed on the path-augmented supervoxel graph.

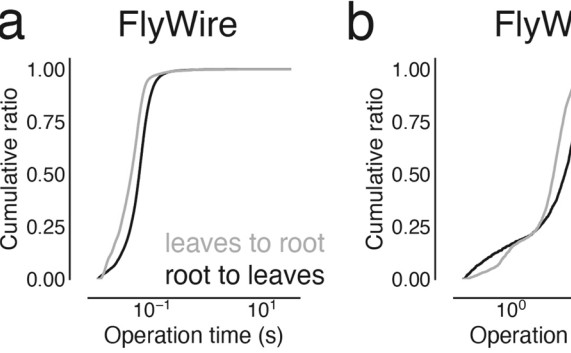

**Extended Data Fig. 2 | ChunkedGraph performance measurements on FlyWire.** These measurements are from the improved ChunkedGraph implementation using the same FlyWire supervoxel graph that was used for the original implementation[29]. (**a**) Performance measurement from real-world user interactions measured on the ChunkedGraph server for reads, specifically leaves to root (median=41 ms, N = 13,410) and root leaves (median=55 ms, N = 50,001) operations, and (**b**) edits, specifically merge (median=2,734 ms, N = 4,189) and split (median=3,486 ms, N = 2,875) operations. The cumulative ratio of all measured interactions for a given response time is plotted in the y axis.

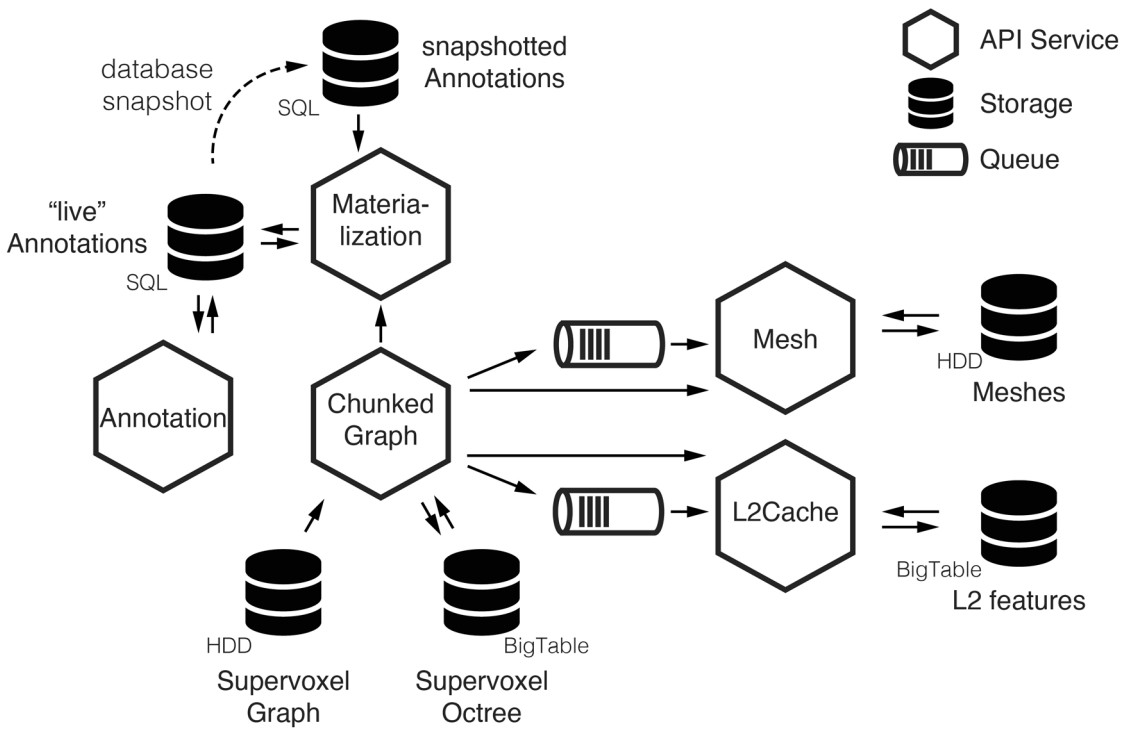

**Extended Data Fig. 3 | Overview of the core CAVE services, their storage and interactions.** Arrows indicate flow of data between services and storage backends. Services are implemented as microservices deployed through kubernetes.

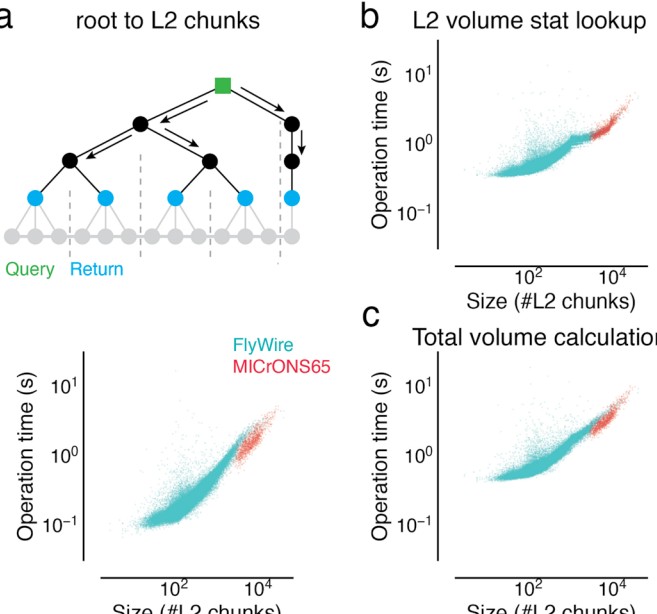

**Extended Data Fig. 4 | Analysis of timings to calculate morphological features.** Each dot is a query for a single neuron. (**a**) Times to retrieve a list of L2 chunks for a neuron (root id). (**b**) Time to look up volume measurements for all L2 chunks belonging to a given neuron. (**c**) Total time to calculate volumes for neurons. Number of samples for all plots: N(FlyWire) = 101,554; N(MICrONS65) = 1,357.

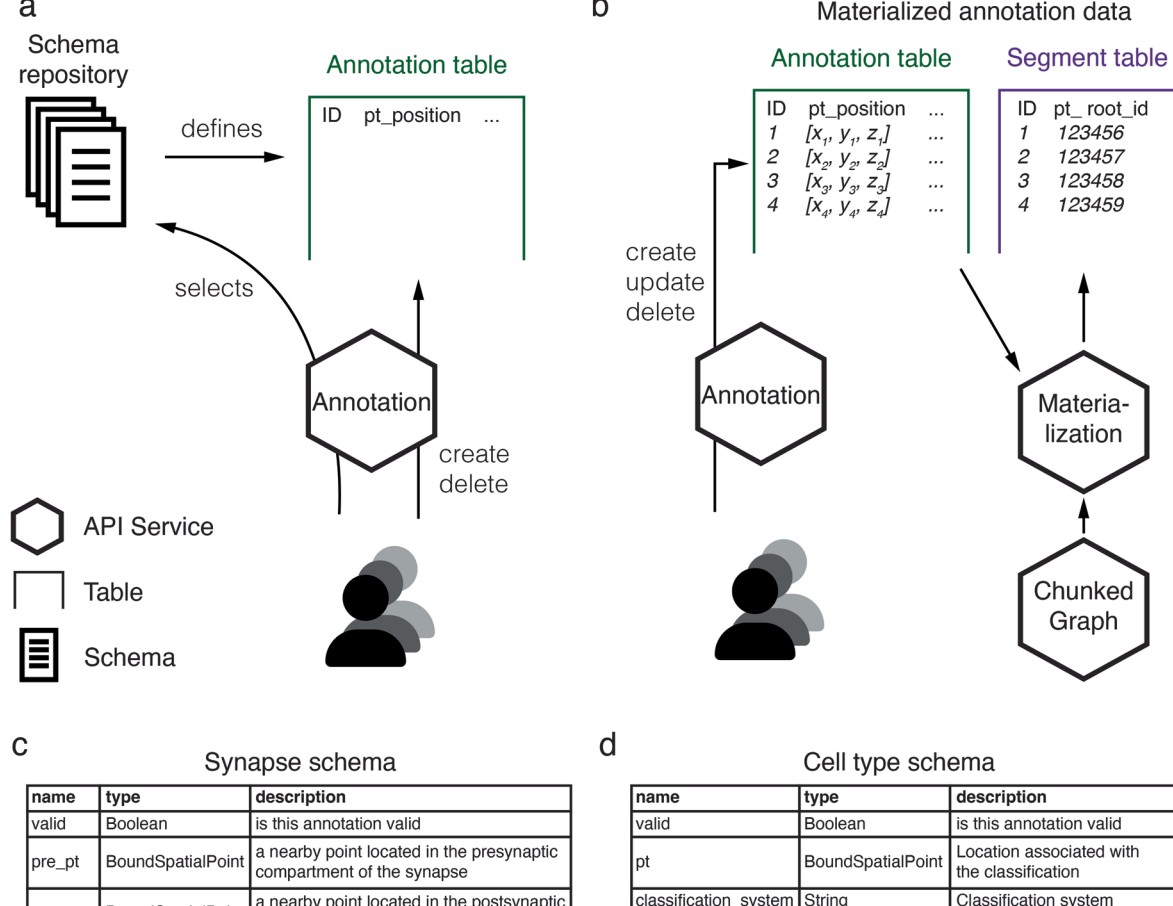

**c** Synapse schema

| name | type | description |
|---|---|---|
| valid | Boolean | is this annotation valid |
| pre_pt | BoundSpatialPoint | a nearby point located in the presynaptic compartment of the synapse |
| post_pt | BoundSpatialPoint | a nearby point located in the postsynaptic compartment of the synapse |
| ctr_pt | SpatialPoint | central point |
| size | Float | size of synapse |

**d** Cell type schema

| name | type | description |
|---|---|---|
| valid | Boolean | is this annotation valid |
| pt | BoundSpatialPoint | Location associated with the classification |
| classification_system | String | Classification system |
| cell_type | String | Cell type name |

**e** Example materialized annotation tables

Annotation table

| ID | valid | pt_position | classification_system | cell_type |
|---|---|---|---|---|
| 338 | t | [187072, 144336, 20909] | aibs_coarse_inhibitory | BC |
| 339 | t | [172512, 175280, 21964] | aibs_coarse_inhibitory | BPC |
| 340 | t | [191728, 157904, 22608] | aibs_coarse_excitatory | 23P |
| 2030 | t | [174432, 200608, 21567] | aibs_coarse_inhibitory | BC |
| 2031 | t | [192384, 216944, 21284] | aibs_coarse_inhibitory | BC |

Segment table

| ID | pt_supervoxel_id | pt_root_id |
|---|---|---|
| 338 | 90505407025912875 | 864691136388711031 |
| 339 | 88468836747612860 | 864691135654475970 |
| 340 | 91140581417423431 | 864691135731235769 |
| 2030 | 88753747697717397 | 864691135211359680 |
| 2031 | 91218852700367385 | 864691134948652540 |

**Extended Data Fig. 5 | Schematic of annotation databases, schemas, and annotation tables.** (**a**) Users can create and delete annotation tables through the annotation service. When creating a table, users select one of many available schemas that define the columns in the annotation table. (**b**) Users can create, update and delete annotations in the annotation table. The materialization service then adds these annotations to the associate segment table and regularly updates the root ids (that is, segment ids) associated with these annotations. (**c**) A commonly used schema for synapses. Each row defines a column in the annotation table. Entries of type BoundSpatialPoint are linked to the underlying segmentation and updated by the materialization service in the segment table. (**d**) Same as (**c**) but for a cell type schema. (**e**) Examples from an annotation table using the cell type schema from (**d**) in the MICrONS dataset.

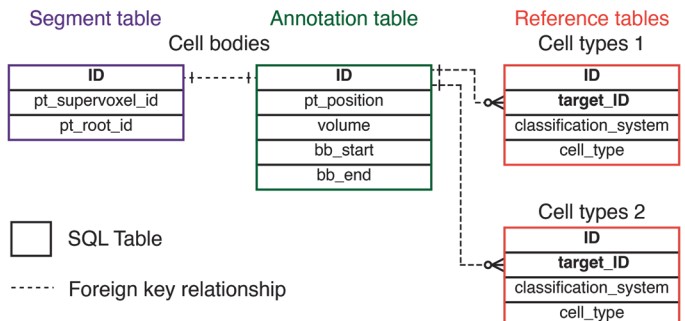

**Extended Data Fig. 6 | Foreign key relationships between tables.** This example shows how annotation and segment tables for nucleus annotations are combined and further extended with reference tables. Annotation and segment tables are automatically combined by the Materialization service via a foreign key relationship on their ID columns. Reference tables created by the user also use foreign key relationships to associate additional information with rows in an annotation table. Multiple such reference tables can point at one Annotation table.

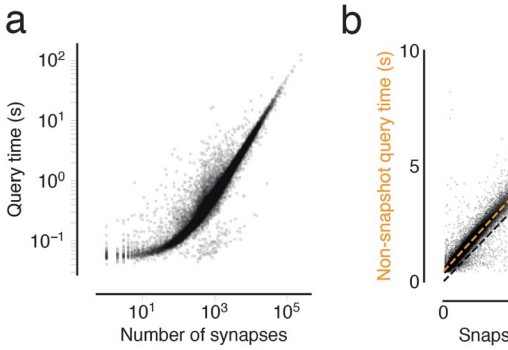

**Extended Data Fig. 7 | Annotation query timing analysis. (a)** Query times from Fig. 5d versus the size of the query in number of presynapses (N = 121,400). (**b**) Comparing snapshot and non-snapshot aligned presynapse queries for cases where the neuron was not edited between the snapshot and the query time (N = 121,367). The difference is the overhead of the mapping logic. The orange dashed line is a linear fit with intercept 0.44 s and a slope of 1.05.

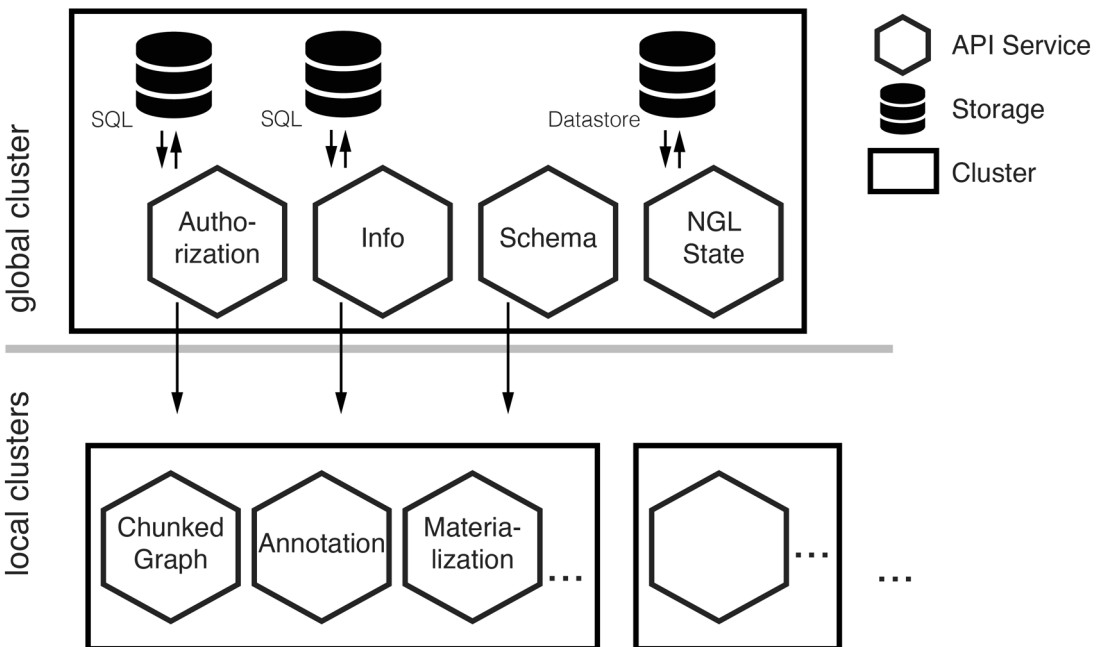

**Extended Data Fig. 8 | Microservice organization into global and local clusters.** CAVE splits services into global and local clusters dependent on their function. Services in the global cluster are light-weight and usually support a wide array of datasets, provide general information about datasets, and user level authentication. Services on local clusters may require more resources and might be specific to a few datasets. Local clusters are usually limited to a list of datasets they can service and are associated with a global cluster. Multiple local clusters can be associated with the same global cluster.

# Reporting Summary

## Statistics

For all statistical analyses, confirm that the following items are present in the figure legend, table legend, main text, or Methods section.

| n/a | Confirmed | |
|---|---|---|
| ☐ | ☒ | The exact sample size (*n*) for each experimental group/condition, given as a discrete number and unit of measurement |
| ☐ | ☒ | A statement on whether measurements were taken from distinct samples or whether the same sample was measured repeatedly |
| ☒ | ☐ | The statistical test(s) used AND whether they are one- or two-sided *Only common tests should be described solely by name; describe more complex techniques in the Methods section.* |
| ☒ | ☐ | A description of all covariates tested |
| ☒ | ☐ | A description of any assumptions or corrections, such as tests of normality and adjustment for multiple comparisons |
| ☐ | ☒ | A full description of the statistical parameters including central tendency (e.g. means) or other basic estimates (e.g. regression coefficient) AND variation (e.g. standard deviation) or associated estimates of uncertainty (e.g. confidence intervals) |
| ☒ | ☐ | For null hypothesis testing, the test statistic (e.g. *F*, *t*, *r*) with confidence intervals, effect sizes, degrees of freedom and *P* value noted *Give P values as exact values whenever suitable.* |
| ☒ | ☐ | For Bayesian analysis, information on the choice of priors and Markov chain Monte Carlo settings |
| ☒ | ☐ | For hierarchical and complex designs, identification of the appropriate level for tests and full reporting of outcomes |
| ☒ | ☐ | Estimates of effect sizes (e.g. Cohen's *d*, Pearson's *r*), indicating how they were calculated |

*Our web collection on statistics for biologists contains articles on many of the points above.*

## Software and code

Policy information about availability of computer code

| | |
|---|---|
| Data collection | The raw data used in this study where collected by previous consortium efforts (ref. 14, ref. 16) and are publicly available. CAVE was used by these efforts to proofread and annotate these datasets. Because each project spanned multiple years, several versions of CAVE-related packages were used as they were improved over time. For every CAVE-related package (as listed in Supplementary Table 1) the history of it's versions is available on GitHub. Most relevant to the performance analyses in this study is the distinction of the major ChunkedGraph versions. Here, we introduced and analyzed the version 2 of the ChunkedGraph and compare it to the version 1 that was published and described previously. |
| Data analysis | Data analysis in this study was mostly limited to collecting and plotting data extracted from CAVE deployments for MICrONS and FlyWire. We provide a github repository with notebooks to reproduce all Figures in the paper at https://github.com/CAVEconnectome/CAVEpaper |

For manuscripts utilizing custom algorithms or software that are central to the research but not yet described in published literature, software must be made available to editors and reviewers. We strongly encourage code deposition in a community repository (e.g. GitHub). See the Nature Portfolio guidelines for submitting code & software for further information.

## Data

Policy information about availability of data

All manuscripts must include a data availability statement. This statement should provide the following information, where applicable:
- Accession codes, unique identifiers, or web links for publicly available datasets
- A description of any restrictions on data availability
- For clinical datasets or third party data, please ensure that the statement adheres to our policy

> The datasets used in this manuscript are publicly available. The MICrONS dataset (ref. 14) can be accessed at microns-explorer.org/cortical-mm3 and https://bossdb.org/project/microns-minnie. The FlyWire dataset (ref. 16) can be accessed at flywire.ai.

## Human research participants

Policy information about studies involving human research participants and Sex and Gender in Research.

| | |
|---|---|
| Reporting on sex and gender | n/a |
| Population characteristics | n/a |
| Recruitment | n/a |
| Ethics oversight | n/a |

Note that full information on the approval of the study protocol must also be provided in the manuscript.

# Field-specific reporting

Please select the one below that is the best fit for your research. If you are not sure, read the appropriate sections before making your selection.

☒ Life sciences ☐ Behavioural & social sciences ☐ Ecological, evolutionary & environmental sciences

For a reference copy of the document with all sections, see nature.com/documents/nr-reporting-summary-flat.pdf

# Life sciences study design

All studies must disclose on these points even when the disclosure is negative.

| | |
|---|---|
| Sample size | We evaluated CAVE on two datasets. For each anak |
| Data exclusions | No data was excluded. |
| Replication | We performed most analyses once per dataset. Most analyses were performed in both datasets. Analyses of the annotation query times where only performed in the FlyWire dataset (Fig. 5) and not replicated in the MICrONS dataset due to the much higher number of measurements available in the FlyWire dataset. |
| Randomization | No randomization was involved in this study. Groups are not being compared and so randomization was not relevant. |
| Blinding | Blinding was not applicable as no group assignments were performed. |

# Reporting for specific materials, systems and methods

We require information from authors about some types of materials, experimental systems and methods used in many studies. Here, indicate whether each material, system or method listed is relevant to your study. If you are not sure if a list item applies to your research, read the appropriate section before selecting a response.

## Materials & experimental systems

| n/a | Involved in the study |
|-----|----------------------|
| ☒ | Antibodies |
| ☒ | Eukaryotic cell lines |
| ☒ | Palaeontology and archaeology |
| ☒ | Animals and other organisms |
| ☒ | Clinical data |
| ☒ | Dual use research of concern |

## Methods

| n/a | Involved in the study |
|-----|----------------------|
| ☒ | ChIP-seq |
| ☒ | Flow cytometry |
| ☒ | MRI-based neuroimaging |

