## [Peer Review File · Nature Methods]

Peer Review Information

Manuscript Title: CAVE: Connectome Annotation Versioning Engine

Corresponding author name(s): Forrest Collman

Editorial Notes:

Transferred manuscripts This manuscript has been previously reviewed at another journal that is not operating a transparent peer review scheme. This document only contains reviewer comments, rebuttal and decision letters for versions considered at *Nature Methods*.

Reviewer Comments & Decisions:

Decision Letter, initial version:

Dear Forrest,

Thank you for submitting your revised manuscript "CAVE: Connectome Annotation Versioning Engine" (NMETH-A54507-T). It has now been seen by one of the original referees and their comments are below. The reviewer find that the paper has improved in revision, and therefore we'll be happy in principle to publish it in Nature Methods, pending minor revisions to satisfy the referees' final requests and to comply with our editorial and formatting guidelines.

TRANSPARENT PEER REVIEW

Please note: we allow redactions to authors' rebuttal and reviewer comments in the interest of confidentiality. If you are concerned about the release of confidential data, please let us know

specifically what information you would like to have removed. Please note that we cannot incorporate redactions for any other reasons. Reviewer names will be published in the peer review files if the reviewer signed the comments to authors, or if reviewers explicitly agree to release their name. For more information, please refer to our FAQ page.

ORCID

Best regards,
Nina

Nina Vogt, PhD
Senior Editor
Nature Methods

Reviewer #3 (Remarks to the Author):

This revised manuscript largely addresses my previous comments and suggestions. The additions to the Discussion help place the work in context and address future scalability. The new extended data figures will be useful to readers seeking a deeper understanding of the design architecture and preconfigured schemas. The additional methods section "CAVE deployment and cost" will help readers understand pragmatically how CAVE is positioned in the larger ecosystem of tools one can apply to connectomes (and related) problems. This manuscript now successfully conveys the state-of-the-art in software analysis infrastructure for large-scale connectomics datasets.

A few comments and suggested improvements do arise in response to the revisions:

- 1) On line 492, "The ChunkedGraph will need to implement supervoxel splitting to allow supervoxels to be grown in size and reduce their number." -- this sentence was unclear. How does splitting supervoxels allow them to grow in size and reduce in number?
- 2) On line 498, "new storage solutions will be needed" -- any prospects for this, or is it more of a generic hope? For example, does BigTable continue to be the core, or are any open-source solutions available (or forthcoming) that could replace it? Within closed-source solutions, should BigTable (which I understand to be better for reads than writes) be replaced by something like Spanner (which I understand has better read/write performance)?
- 3) Ext data 4-1 is very useful but the authors might also consider including a diagram showing relationships between tables schema, i.e. foreign key relationships across tables, in the mode of a more traditional relational database schema. (I believe this is the sort of thing the other reviewer may have been looking for as well.) Illustrating at least part of the inter-table schema for the worked

example of the FlyWire project could be very helpful here.

4) (minor) Consider lightly colorizing Ext Data 3-1 to help different component types stand out from one another

5) (typo) Figure 4e Legend "Illustration on how"  "Illustration of how"

Author Rebuttal to Initial comments

Reviewer 3

This revised manuscript largely addresses my previous comments and suggestions. The additions to the Discussion help place the work in context and address future scalability. The new extended data figures will be useful to readers seeking a deeper understanding of the design architecture and preconfigured schemas. The additional methods section "CAVE deployment and cost" will help readers understand pragmatically how CAVE is positioned in the larger ecosystem of tools one can apply to connectomes (and related) problems. This manuscript now successfully conveys the state-of-the-art in software analysis infrastructure for large-scale connectomics datasets.

A few comments and suggested improvements do arise in response to the revisions:

3.1) On line 492, "The ChunkedGraph will need to implement supervoxel splitting to allow supervoxels to be grown in size and reduce their number." -- this sentence was unclear. How does splitting supervoxels allow them to grow in size and reduce in number?

We regret the sentence was not clear, in fact the logic is inverse of what was stated. Growing supervoxels in size and reducing their number will mean that some supervoxels will require splitting in order to make the segmentation accurate. Therefore, the likely easiest strategy to reduce costs as the ChunkedGraph scales is to grow supervoxels size and implement supervoxel splitting to alleviate the problems created when supervoxels are in fact no longer immutable. We have added a sentence in hopes of clarifying this point.

3.2) On line 498, "new storage solutions will be needed" -- any prospects for this, or is it more of a generic hope? For example, does BigTable continue to be the core, or are any open-source solutions available (or forthcoming) that could replace it? Within closed-source solutions, should BigTable (which I understand to be better for reads than writes) be replaced by something like Spanner (which I understand has better read/write performance)?

This sentence was focused mostly on annotations, and not the segmentation. BigTable is not directly used in storing such annotations. The best prospect we see for reducing the costs of storing large scale annotation data is to move it toward lower cost storage solutions and reduce data duplication. Similar to image and mesh data, we likely need most static data to be delivered by serverless technologies that are backed by cheaper storage options, while having sufficient indexing to make retrieval of those data efficient for the most common use cases. Our present implementation makes snapshots of databases to use as a cache, but in the process creates data duplication, and further stores most data on relatively expensive spinning hard disks attached to a Postgres server. Future solutions should segregate the static and dynamic portions of these data in different storage tiers. We think this is possible using technologies like Occasionally-cooperative distributed b-trees (OCDBT by Tensorstore), which will provide a key value store for mostly static data with flexible b-tree indexes to enable efficient retrieval that is backed directly by lower cost cloud storage, with no additional server costs.

We have modified the sentence to try to more clearly highlight the specific directions that we think most promising, but given that we haven't yet solved it we didn't want to be overly prescriptive in how this will happen.

3.3) Ext data 4-1 is very useful but the authors might also consider including a diagram showing relationships between tables schema, i.e. foreign key relationships across tables, in the mode of a more traditional relational database schema. (I believe this is the sort of thing the other reviewer may have been looking for as well.) Illustrating at least part of the inter-table schema for the worked example of the FlyWire project could be very helpful here.

In our database scheme, the only relationships that exist between annotations are based on reference annotations. This is tied fundamentally to the idea that users are free to create new tables of a given schema dynamically and that the infrastructure supports an expanding set of schemas to cover any potential use case. We have taken the suggestion and added a standard relationship diagram to illustrate concrete tables currently used in flywire and MICrONS, including reference annotation tables.

3.4) (minor) Consider lightly colorizing Ext Data 3-1 to help different component types stand out from one another

We appreciate the suggestion to make this Extended Data Figure more clear. However, we think the current symbols are sufficient to differentiate the component types and keep the figure uncolored for coherence across figures.

3.5) (typo) Figure 4e Legend "Illustration on how"  "Illustration of how"

Thanks for this catch. Fixed.

Final Decision Letter:

Dear Forrest,

I am pleased to inform you that your Article, "CAVE: Connectome Annotation Versioning Engine", has now been accepted for publication in Nature Methods. The received and accepted dates will be February 6th, 2024 and August 19th, 2024. This note is intended to let you know what to expect from us over the next month or so, and to let you know where to address any further questions.

Over the next few weeks, your paper will be copyedited to ensure that it conforms to Nature Methods style. Once your paper is typeset, you will receive an email with a link to choose the appropriate publishing options for your paper and our Author Services team will be in touch regarding any additional information that may be required. It is extremely important that you let us know now whether you will be difficult to contact over the next month. If this is the case, we ask that you send us the contact information (email, phone and fax) of someone who will be able to check the proofs and deal with any last-minute problems.

Please note that *Nature Methods* is a Transformative Journal (TJ). Authors may publish their research with us through the traditional subscription access route or make their paper immediately open access through payment of an article-processing charge (APC). Authors will not be required to make a final decision about access to their article until it has been accepted. Find out more about Transformative Journals

Authors may need to take specific actions to achieve compliance with funder and institutional open access mandates. If your research is supported by a funder that requires immediate open access (e.g. according to Plan S principles) then you should select the gold OA route, and we will direct you to the compliant route where possible. For authors selecting the subscription publication route, the journal's standard licensing terms will need to be accepted, including self-

archiving policies. Those licensing terms will supersede any other terms that the author or any third party may assert apply to any version of the manuscript.

If you are active on Twitter/X, please e-mail me your and your coauthors' handles so that we may tag you when the paper is published.

Best regards,
Nina

Nina Vogt, PhD
Senior Editor
Nature Methods